# Treatment Outcomes of Adolescents Compared to Younger Pediatric Patients with Acute Myeloid Leukemia: Do They Need a Special Approach?

**DOI:** 10.3390/cancers16061145

**Published:** 2024-03-14

**Authors:** Katarzyna Pawińska-Wąsikowska, Małgorzata Czogała, Karolina Bukowska-Strakova, Marta Surman, Monika Rygielska, Teofila Książek, Beata Sadowska, Agnieszka Pac, Jolanta Skalska-Sadowska, Magdalena Samborska, Jacek Wachowiak, Małgorzata Ciebiera, Radosław Chaber, Renata Tomaszewska, Tomasz Szczepański, Karolina Zielezińska, Tomasz Urasiński, Małgorzata Moj-Hackemer, Krzysztof Kałwak, Marta Kozłowska, Ninela Irga-Jaworska, Barbara Sikorska-Fic, Paweł Łaguna, Katarzyna Muszyńska-Rosłan, Maryna Krawczuk-Rybak, Anna Fałkowska, Katarzyna Drabko, Katarzyna Bobeff, Wojciech Młynarski, Agnieszka Chodała-Grzywacz, Grażyna Karolczyk, Katarzyna Mycko, Wanda Badowska, Natalia Bartoszewicz, Jan Styczyński, Katarzyna Machnik, Agnieszka Mizia-Malarz, Walentyna Balwierz, Szymon Skoczeń

**Affiliations:** 1Department of Pediatric Oncology and Hematology, Institute of Pediatrics, Jagiellonian University Medical College, 30-663 Krakow, Poland; malgorzata.czogala@uj.edu.pl (M.C.); walentyna.balwierz@uj.edu.pl (W.B.); szymon.skoczen@uj.edu.pl (S.S.); 2Department of Pediatric Oncology and Hematology, University Children’s Hospital, 30-663 Krakow, Poland; 3Department of Clinical Immunology, Institute of Pediatrics, Jagiellonian University Medical College, 30-663 Krakow, Poland; k.bukowska-strakova@uj.edu.pl (K.B.-S.); msurman@usdk.pl (M.S.); 4Department of Pediatric Oncology and Hematology, Hematology Laboratory, University Children’s Hospital, 30-663 Krakow, Poland; mrygielska@usdk.pl; 5Department of Medical Genetics, Institute of Pediatrics, Jagiellonian University Medical College, 30-663 Krakow, Poland; teofila.ksiazek@uj.edu.pl; 6Department of Pediatric Oncology and Hematology, Cytogenetics and Molecular Genetics Laboratory, University Children’s Hospital, 30-663 Krakow, Poland; bsadowska@usdk.pl; 7Department of Epidemiology and Preventive Medicine, Faculty of Medicine, Jagiellonian University Medical College, 31-034 Krakow, Poland; agnieszka.pac@uj.edu.pl; 8Department of Pediatric Oncology, Hematology and Transplantology, Poznan University of Medical Sciences, 60-572 Poznan, Poland; jsk@poczta.onet.eu (J.S.-S.); samborska.magda@gmail.com (M.S.); wachowiak.jacek@outlook.com (J.W.); 9Department of Pediatric Oncohematology, Clinical Province Hospital of Rzeszow, 35-301 Rzeszow, Poland; malgorzata.ciebiera@gmail.com (M.C.); radoslaw.chaber@gmail.com (R.C.); 10Department of Pediatrics, Institute of Medical Sciences, Medical College, University of Rzeszow, 35-310 Rzeszow, Poland; 11Department of Pediatric Hematology and Oncology, Medical University of Silesia, 40-055 Katowice, Poland; rtomaszewska@szpital.zabrze.pl (R.T.); szczep57@poczta.onet.pl (T.S.); 12Department of Pediatrics, Hemato-Oncology and Gastroenterology, Pomeranian Medical University in Szczecin, 71-252 Szczecin, Poland; karolina.zielezinska@pum.edu.pl (K.Z.); urasin@pum.edu.pl (T.U.); 13Clinical Department of Pediatric Bone Marrow Transplantation, Oncology and Hematology, Wroclaw Medical University, 213 Borowska St., 50-556 Wroclaw, Poland; malgorzata.moj@o2.pl (M.M.-H.); krzysztof.kalwak@gmail.com (K.K.); 14Department of Pediatrics, Hematology and Oncology, Medical University of Gdansk, 80-210 Gdansk, Poland; marta.kozlowska@gumed.edu.pl (M.K.); nirga@gumed.edu.pl (N.I.-J.); 15Department of Oncology, Pediatric Hematology, Transplantology and Pediatrics, Medical University of Warsaw, 02-091 Warsaw, Poland; basiasf@poczta.onet.pl (B.S.-F.); pawel.laguna@wum.edu.pl (P.Ł.); 16Department of Pediatric Oncology and Hematology, Medical University of Bialystok, 15-276 Bialystok, Poland; katarzyna.muszynska-roslan@umb.edu.pl (K.M.-R.); rybak@umb.edu.pl (M.K.-R.); 17Department of Pediatric Hematology, Oncology and Transplantology, Medical University of Lublin, 20-090 Lublin, Poland; ania1589@gmail.com (A.F.); katarzynadrabko@umlub.pl (K.D.); 18Department of Pediatrics, Oncology and Hematology, Medical University of Lodz, 91-738 Lodz, Poland; katarzynabobeff@gmail.com (K.B.); wojciech.mlynarski@umed.lodz.pl (W.M.); 19Department of Pediatric Hematology and Oncology, Regional Polyclinic Hospital in Kielce, 25-736 Kielce, Poland; aga.chodala@vp.pl (A.C.-G.); grazyna.karolczyk@wszzkielce.pl (G.K.); 20Department of Pediatrics and Hematology and Oncology, Province Children’s Hospital, 10-561 Olsztyn, Poland; katarzynamycko@o2.pl (K.M.); hematologia@wssd.olsztyn.pl (W.B.); 21Department of Pediatric Hematology and Oncology, Collegium Medicum, Nicolaus Copernicus University Torun, 85-094 Bydgoszcz, Poland; natalabar@wp.pl (N.B.); jstyczynski@cm.umk.pl (J.S.); 22Department of Pediatrics, Hematology and Oncology, City Hospital, 41-500 Chorzow, Poland; katmachnik@gmail.com; 23Department of Pediatrics, Medical University of Silesia, Upper Silesia Children’s Care Health Centre, 40-752 Katowice, Poland; a.mizia@wp.pl

**Keywords:** acute myeloid leukemia, adolescents, young adults, children, treatment-related mortality

## Abstract

**Simple Summary:**

The aim of this study was to retrospectively analyze the characteristics and treatment outcomes of adolescents with AML compared to children and infants treated, according to the two consecutive BFM protocols. It showed that overall survival did not differ significantly between adolescents and children aged 1 to 15 years. However, relapse-free survival was shorter in adolescents compared to younger children, and treatment-related mortality tended to be higher. High-risk genetics and a leukocyte count above 100,000/μL at diagnosis, rather than age above 15 years, proved to be unfavorable prognostic factors for the treatment outcome.

**Abstract:**

Background: The reports of studies that compare the survival of adolescents and young adults with younger children with acute myeloid leukemia (AML) are contradictory. Patients and Methods: We retrospectively analyzed 220 AML patients aged 0–18 years treated in pediatric oncologic centers in Poland from 2015 to 2022. The evaluated group included 31 infants (below 1 year), 91 younger children (1–9.9 years), 59 older children (10–14.9 years), and 39 adolescents (15–18 years). Results: A 5-year overall survival for adolescents was not significantly inferior compared to younger and older children (74.3 ± 7.6% vs. 80.5 ± 4.4% vs. 77.9 ± 5.1, *p* = 0.243). However, relapse-free survival was lower in adolescents compared to younger children (76.5 ± 7.8% vs. 65.7 ± 9.0%, *p* = 0.049), and treatment-related mortality tended to be higher (10.3% vs. 4.4%, *p* = 0.569). In the univariate analysis, high-risk genetics [HR, 2.0 (95% CI 1.1–3.6; *p* = 0.014)] and a leukocyte count at diagnosis above 100,000/μL [HR, 2.4 (95% CI 1.3–4.6; *p* = 0.004)] were found to be unfavorable prognostic factors for survival. Conclusions: Although we have not found that age over 15 years is an unfavorable factor for overall survival, the optimal approach to therapy in adolescents, as in other age groups, is to adjust the intensity of therapy to individual genetic risk and introduce targeted therapies when indicated.

## 1. Introduction

The outcomes in pediatric acute myeloid leukemia (AML) have improved considerably in recent years. Long-term survival has reached 70–80%, mainly due to advances in chemotherapies, hematopoietic stem cell transplantation (HSCT), supportive care, and the implementation of targeted therapies, such as FLT3-ITD inhibitors. Survival improved regardless of age; however, treatment outcomes for adolescents and young adults are still worse compared to younger children, but superior to middle-aged and older adults [1,2,3,4,5,6,7]. Age at diagnosis is still considered one of the crucial prognostic factors for acute leukemias [8,9].

Adolescents and young adults (AYA) are often defined as patients aged 15 to 39 years, creating a unique group of patients from a pediatric and adult setting, treated according to pediatric or adult treatment protocols [10,11]. Adolescents are in the so-called ‘in-between’ age group with a special toxicity profile, psychosocial needs, and generally a lower compliance rate [12,13,14].

The results of recent clinical trials regarding the impact of age on survival rates in AYA with AML are contradictory. In most studies, the survival of AYA was lower compared to that of children. Reports from St. Jude Children’s Research Hospital, Children’s Oncology Group (COG), Cancer and Leukemia Group B (CALBG), and SWOG (Southwest Oncology Group) studies showed that overall survival (OS) and event-free survival (EFS) were similar between younger patients and AYA patients, although the latter had higher treatment-related toxicity (TRM) [15,16,17]. It remains in contrast to the results of other studies. In the BFM study, OS and EFS were significantly lower for adolescents and young adults, with no differences in TRM across all ages [18]. Furthermore, in a nationwide population study of Dutch patients with AML, AYA showed worse OS compared to children, but early mortality did not differ between children and AYA patients [19].

In Poland, patients are treated according to pediatric protocols up to 18 years of age. Therefore, we decided to evaluate the results of adolescents aged 15 to 18 years to answer the question of whether this unique group of patients needed a special treatment approach. The objective of this study was to retrospectively evaluate the characteristics and long-term survival of adolescents with acute myeloid leukemia compared to children and infants treated according to the two consecutive BFM Study Group protocols in Poland.

## 2. Materials and Methods

We retrospectively analyzed the AML database of the Polish Pediatric Leukemia and Lymphoma Study Group. There were 314 pediatric de novo patients with AML registered between 2015 and 2022. Patients with acute promyelocytic leukemia (n = 18), myeloid leukemia in Down syndrome (n = 33), myelodysplastic syndrome (MDS)-related AML (n = 18), AML after cytotoxic therapy (n = 17) and mixed lineage phenotype leukemia—MPAL (n = 8) were excluded from the analysis.

We eventually evaluated 220 patients with AML aged 0–18 years. The patients were grouped into four cohorts by age at diagnosis: infants, less than 1 year old (0–0.9 year); younger children, aged 1 to less than 10 years old (1–9.9 years); older children, aged 10 to less than 15 years old (10–14.9 years); and patients aged 15 to 18 years were considered adolescents.

From 2015–2019, patients were treated with the AML-BFM 2012 Registry (131 patients) and from 2019–2022 with the AML-BFM 2019 Recommendations (89 patients) protocols. The median follow-up time was 49 (0.2–61.8) months. The protocols were approved by the respective institutional review boards and written informed consent was obtained from all patients.

In this study, we distinguish three groups of patients in terms of genetic alterations in leukemic cells. The low-risk genetics (abnormal genetics with favorable prognosis) group consists of patients with t(8;21)(q22;q22)/RUNX1::RUNX1T1 and inv(16)(p13.1q22) or t(16;16)(p13.1;q22)/CBFB::MYH11; high-risk genetics (abnormal genetics with unfavorable prognosis) consist of patients with monosomy 7, t(6;9)(p23;q34)/DEK::NUP214, t(9;22)(q34;q11.2)/BCR::ABL1; FLT3-ITD and mutated WT1, or 11q23/KMT2A abnormalities, i.e., t(6;11)(q27;q23)/KMT2A::AF6; t(10;11)(p12;q23)/KMT2A::MLLT10; or complex karyotypes defined as three or more aberrations, including at least one structural aberration, without favorable genetics and without KMT2A rearrangement in leukemic cells. The rest of the patients with the absence of low or high-risk genetic features were classified into the intermediate-risk genetic group. Patients with normal karyotypes were classified as the intermediate-risk group. The normal karyotype was defined by the absence of clonal aberration in the metaphases of bone marrow aspirates at the time of diagnosis.

We used standard, generally accepted definitions for response criteria and treatment failures. The definition of complete remission (CR) was described as less than 5% of the blasts in bone marrow aspirate smears, with an absolute neutrophil count >1 × 10^9^/L, platelets 100 × 10^9^/L in peripheral blood, and no signs of extramedullary involvement. Primary refractory disease was defined as more than 10% blasts after the first induction, assessed between days 28 and 42 since the start of the induction, and/or >5% blasts after the second induction block. Relapse was defined as the reappearance of more than 5% leukemic blasts in a representative bone marrow identified by microscopic and/or flow cytometry methods and/or evidence of leukemic infiltration of any site. The time of relapse defines the types of relapse. Early relapse was diagnosed within 18 months after initial diagnosis, while relapse after 18 months of diagnosis was considered late. Early death was defined as the death of any cause that occurred within 42 days after initial diagnosis. Death of the patient at the time of remission or absence of progression was defined as treatment-related death.

Patients were observed from the date of AML diagnosis until death, loss of follow-up, or 1 August 2023 (end of follow-up), whichever comes first.

The study protocol has been carried out in accordance with The World Medical Association Ethics Code (Declaration of Helsinki) for experiments involving humans and approved by the Ethics Committee of Jagiellonian University (protocol code—118.6120.24.2023, date of approval—15 June 2023).

### 2.1. Treatment

Age was not a factor in the assignment of the risk group in either of the two treatment protocols. In the AML-BFM 2012 Registry and AML-BFM 2019 protocols after two courses of induction chemotherapy once complete remission was attained, patients were stratified into one of the risk groups, standard, intermediate or high, mainly based on genetic abnormalities of leukemic cells and response to therapy. Consolidation treatment consisted of two or three chemotherapy courses, depending on the risk group. Patients stratified into the high-risk group received allogeneic hematopoietic stem cell transplantation (allo-HSCT), whereas non-high-risk patients received one year of maintenance therapy. All patients received standard-of-care posaconazole or voriconazole for the prevention of fungal infection and trimethoprim/sulphamethoxazole (TMP/SMX) as a prophylaxis of *Pneumocystis jirovecii* infection. Details of both treatment protocols and stratification to risk group according to treatment protocols.are given in the Appendix A.

### 2.2. Statistical Analysis

We used the Kaplan-Meier method to estimate the survival probability of our patients. The main endpoints of the study were event-free survival (EFS), relapse-free survival (RFS), cumulative incidence of relapse (CIR), and overall survival (OS). The EFS was calculated as the time from the date of leukemia diagnosis to the first failure event, such as disease progression, relapse, or death. The last follow-up was used as a censored date for patients with no failure event. The RFS and CIR were defined as the time from the date of complete remission to the date of relapse. CIR was used to estimate the rate of relapses. OS was calculated as the time from the date of initial diagnosis to death from any cause. Treatment-related mortality (TRM) was defined as death from non-progressive disease.

Patients who have not died were censored at their last follow-up date. Survival probabilities were presented along with standard errors (SE). Variables that include age at initial diagnosis (infants, 0–1 year old; younger children, 1–9 years old; older children, 10–14 years old; and patients 15 years old and older), leukocyte count at initial diagnosis, sex, treatment risk groups and presence of high-risk genetics, absence of low-risk genetics, allo-HSCT procedure were analyzed. The subgroups were compared with a log-rank test. A Cox regression model was used to identify the risk factors associated with the EFS and OS rates. Variables including age at diagnosis (<1 year vs. >1 year and <15 years vs. >15 years), white blood cell counts at diagnosis (≥100,000/μL vs. <100,000/μL), and the presence of high-risk genetics were analyzed. The significant variables associated with OS, EFS, and TRM were searched.

The significance level of 0.05 was used in all statistical tests. Statistical analyses were performed with SPSS (Statistical Package for Social Science) version 29.0.

## 3. Results

### 3.1. Patient’s Characteristics

Among 220 children evaluated in the presented study, there were 31 patients in the infant group (age 0–0.9 year), 91 in the younger children group (1–9.9 years), 59 in the older children group (10–14.9 years), and 39 in the adolescent group (15–18 years). The median age at the time of diagnosis for adolescents was 16.4 years (range: 15.1–17.8 years). There were more girls in this age group (25 girls vs. 14 boys). No differences were found between the four age groups in terms of sex and leukocyte count at diagnosis. Hyperleukocytosis (WBC at diagnosis above 100,000/μL) was shown in 26% of adolescents, 27% of older children, 22.5% of infants, and 14% of younger children.

Most of the infants were diagnosed with FAB M5 (n = 16; 51.6%), while in adolescents, FAB M2 and M4 were the most common (12 patients, 30.8%, each). In the oldest age group, most patients were stratified into a standard-risk group, SRG (15 patients, 39%), mainly due to low-risk genetics, 11 patients (28%) into an intermediate-risk group (IRG), and 13 (33%) to a high-risk group (HRG).

The favorable genetics were often reported in adolescents: 10 of 39 (26%) had low-risk genotype at diagnosis, mainly inv(16)(p13;q22) in cytogenetics, confirmed by the presence of the fusion gene CFB::MYH11. The proportion of core binding factor leukemia (CBF) leukemia, characterized by the presence of either t(8;21)(q22;q22) or inv(16)(p13.1q22) or t(16;16)(p13.1;q22) was significantly higher in adolescents (15 of 39, 38%) (*p* = 0.01). In the older children, aged 10–14.9 years, 20% of patients (12 of 59) and 21% of younger children (19 of 91), aged 1–9.9 were diagnosed with CBF AML. A higher incidence of 11q23 abnormalities (12 of 31, 38%) was observed in infants, whereas favorable genetics such as CBF AML characteristics were not reported in this age group. In 3 out of 39 (7.5%) adolescents, internal tandem duplication of the FMS-like tyrosine 3 gene (FLT3-ITD) was found, and comparable incidence was observed in the group of older children (4 of 59, 7%). The normal karyotype was the most frequent in older children (24%) compared with adolescents, younger children, and infants (18% vs. 18% vs. 19%, respectively); however, the differences were not statistically significant, *p* > 0.05.

The number of patients who underwent hematopoietic stem cell transplantation in the first CR was similar in all age groups: 32% in the infants vs. 34% in the younger children vs. 27% in the older children vs. 28% in the adolescents. The characteristics of the patients according to age groups are shown in Table 1, and the distribution of low, intermediate, or high-risk genetics across the age group is presented in Figure 1.

### 3.2. Treatment Outcomes

The complete remission rate for all study patients was 91.8%. The CR rate was the highest in the younger children (95%), and the lowest in the infant group (87%), while among the adolescents and older children, respectively, 89.7% and 89.8% of the patients achieved CR, *p* < 0.001. The CR rate was lower in infants mainly due to the high early death rate (12.9%). The high early death rate was also observed in older children (8.4%). In the adolescent group, only one patient died before 42 days of therapy without CR due to pulmonary leukostasis. The highest rate of non-response to treatment occurred in adolescents (7.6%).

The highest relapse rate was in younger children (20.9%) compared to infants and adolescents (19.4% and 17.9%, respectively), while in older children, recurrence occurred in only 1.6% of patients, *p* < 0.05. In the analysis, the probability of 5-year RFS for the adolescents was 76.5 ± 7.8%, whereas for the older children was 95.7 ± 4.3% and 65.7 ± 9.0% for the younger children. Similarly, CIR was the lowest for older children and the highest for younger ones (4.3 ± 4.3% vs. 34.3 ± 9.0%, respectively), *p* = 0.049, (Figure 2a,b).

There were no significant differences in the 5-year EFS rate across the age (59.5 ± 9.0% for the infant group vs. 62.3 ± 8.6% in the younger children group vs. 75.7 ± 6.1% in the older children group vs. 65.6 ± 7.8% in the adolescents group, *p* = 0.412). The probability of 5-year OS was comparable between adolescents, younger and older children (74.3 ± 7.6% vs. 80.5 ± 4.4% vs. 77.9 ± 5.1%, respectively), and decreased in infants (63.5 ± 8.8%); however, the differences were not statistically significant (Figure 3a,b).

In adolescents, there were fewer deaths (9 patients, 23.1%) compared to infants (11 patients, 35.5%), and more than in younger and older children (16 patients, 17.6% and 12 patients, 20.3%). In the adolescents’ group, one patient died of leucostasis before day 15 of induction therapy, 5 patients died of disease progression, and 4 of treatment-related mortality. TRM was the highest among adolescents (10.3%); however, the differences among groups were not statistically significant, *p* > 0.05. Two out of four adolescent patients whose death was related to treatment toxicities died due to post-HSCT complications (sepsis and sinusoidal obstruction syndrome, SOS). Treatment outcomes are presented in Table 2.

Univariate and multivariate analyses did not show that age over 15 years at diagnosis was associated with inferior outcomes. In the univariate analysis, high-risk genetics [HR, 2.0 (95% CI 1.1–3.6; *p* = 0.014)] and a leukocyte count at diagnosis above 100,000/μL [HR, 2.4 (95% CI 1.3–4.6; *p* = 0.004)] were found to be unfavorable prognostic factors for overall survival (Table 3). Neither the univariate nor the multivariate analysis showed that any of the variables assessed were significantly associated with TRM. Statistical significance was not achieved mainly due to the low number of deaths caused by the toxicity of treatment.

## 4. Discussion

Although there have been significant improvements in survival outcomes in the past 30 years, regardless of the age of the patient at diagnosis, there are still differences in survival rates between children, adolescents, and young adults [5,7,9,18,19,20]. As well described in the literature, age has a strong inverse correlation to treatment outcomes [8,9,12,13,14,21].

In the presented study, the outcome of adolescent treatment compared to the younger age groups (below 15 years old) was shown. We found that adolescents compared to children aged 1 to 14 years had a similar EFS rate, with lower OS, which could be explained by a higher incidence of TMR (10.3%), although the latter did not reach statistical significance. Significantly higher TRM has also been shown in other studies compared to younger children [15,16,17,22]. In our study, EFS was not statistically different between adolescents and younger age groups (adolescents, 65.6 ± 7.8% vs. older children, 75.7 ± 6.1% vs. younger children, 62.3 ± 8.6%). Although there was no statistical difference in survival across the age, the OS rate for adolescents was lower compared to children aged 1 to 14 years (adolescents, 74.3 ± 7.6% vs. older children, 77.9 ± 5.1% vs. younger children, 80.5 ± 4.4%, *p* = 0.243).

Several studies showed similar results, although comparison was difficult due to the different age ranges between the compared groups. Canner et al. presented similar findings and examined the outcomes of AYA patients (16 to 20 years) to younger patients by analyzing data from four consecutive COG trials. They found that OS was not significantly different between AYA and younger patients (49 ± 7% vs. 54 ± 5%). Higher TRM (25 ± 6% vs. 12 ± 2%, *p* < 0.001), and lower relapse rate (30 ± 7% vs. 41 ± 3%, *p* = 0.002) in AYA were also shown in AYA [17]. In a retrospective study combining three pediatric AML trials in Japan Tomizawa et al. compared adolescents (15 to 18 years) with AML with other pediatric age groups, infants (0–1 year) and younger (2–11 years) and older children (12–14 years). Adolescents had a similar EFS rate to younger and older children (55.2% vs. 57.6% vs. 63.2%, respectively, *p* = 0.578), but a lower OS rate (54.7% vs. 73.2% vs. 75.5%, respectively, *p* = 0.019), which was explained by a higher incidence of treatment-related deaths in the adolescent group. Similarly, to our results, the worst treatment outcomes were observed in infants compared to children aged 1 to 14 years, and, however, the probability of 5-year OS was better than for adolescents (EFS, 47.4% and OS, 68.7%) [22]. Another study by Rubnitz et al. from St. Jude Children’s Research Hospital also showed comparable outcomes between older patients (10 to 21 years) and younger patients (0 to 9 years), EFS, 58.3 ± 5.4% vs. 66.6 ± 4.9%, *p* = 0.2; OS, 68.9 ± 5.1% vs. 75.1 ± 4.5%, *p* = 0.36; CIR, 21.9 ± 4.4% vs. 25.3 ± 4.1%, *p* = 0.59. Rubnitz et al. found a significantly higher rate of TRM in patients aged 10 to 21 years, mainly due to infections [15].

Several issues have been raised in the literature that could be potentially responsible for inferior outcomes in AYA with AML. Most of all of the differences in cancer biology decreased with age rate of favorable genetic alterations, treatment-related toxicity including transplant-related mortality, problems with compliance to therapy, treatment delays due to complications, prolonged hematologic regeneration, and multidrug resistance increasing with age [8,12,13,14,21].

Although we did not prove that age over 15 years adversely influences the outcome, our study confirmed that the mutational genotype of leukemic cells is still a significant independent prognostic factor, as well as the leukemic load at diagnosis measured by a leukocyte count. Only high-risk genetics and hyperleukocytosis negatively influenced the outcomes. These results are consistent with earlier findings [17,22]. Canner et al. found in multivariate analyses that patients who are overweight, African American or Hispanic and with high-risk cytogenetic abnormality and WBC above 100,000/μL were associated with poor outcomes [17]. Tomizawa et al. showed that age above 15 years at diagnosis was associated with a poor survival rate and a higher TRM. The same authors also showed that high-risk genetic factors, the lack of low-risk genes, WBC above 100,000/μL, as well as age over 15 years and less than 1 year at diagnosis, were risk factors for poor survival [22]. However, the results of the analysis of the pediatric trials AML-BFM 93/98 and the adult trials AMLCG92/99 and AMLSG HD93/98A showed that only cytogenetic abnormalities, blast count after induction (above 5%) and age groups (above 2 years and less than 21 years) remained of prognostic significance. Leucocyte count above 100,000/μL at diagnosis did not have prognostic value [18].

It is well described that favorable genetic alterations are more common in children and the incidence of an unfavorable genetic profile increases with age, which partly explains the superior outcomes in pediatric AML compared to adults [8,21,23]. In our study, CBF leukemia was more common in the adolescent cohort (38%), while FLT3-ITD mutations, typically higher in adolescents, were reported in 7.5% of patients. The normal karyotype was found in 18% of teenagers. Favorable genetics could have contributed to the low relapse rate in adolescents compared to other age groups (5-year CIR: 23.5 ± 7.8, *p* = 0.049).

It is not clear if only genetic alterations are responsible for poorer outcomes in AYA. Mutations associated with aging and clonal hematopoiesis are known disease drivers, and, usually, they are typical for older adults. Some of them could be used as an aim for the targeted therapies. The therapeutic and prognostic impact of new genetic abnormalities is being continuously evaluated [24,25].

As reported by other groups also in our study, there was a high rate of TRM (10.3%) among adolescents [9,10,11,15,16,17]. Toxic deaths were in part due to infections (two out of four), although prophylactic measures were used in all age groups. The other two were related to complications from HSCT. Importantly, the highest death rate (35.5%) was reported in infants, including the high early death rate (12.9%). In our study, this age group was characterized by the worst treatment results. As shown by Creutzig et al. also in our study, favorable genetics were rarely reported in infants. Most infants were included in the high-risk group based on unfavorable genetics, mainly due to KMT2A abnormalities and poor response to therapies (lowest CR rate and high CIR) [8]. Since other study groups showed similar results, more research should focus on this group of patients, disease biology, and treatment [18,22].

Interestingly, the best survival was shown in the younger children group (1–9.9 years) (5-year OS 80.5 ± 4.4%), despite the highest rate of relapse (5-year CIR 34.3 ± 9.0% and 5-year RFS 65.7 ± 9.0%), proving that children in this age group, even if they experienced relapse, still have a chance to be cured.

Studies in adolescents and young adults with acute lymphoblastic leukemia (ALL) have clearly shown a better outcome using more intense pediatric protocols [26,27]. It is not so clear with respect to AML therapies. Gupta et al., in a population-based study in Canada, showed comparable outcomes of AYA (aged 15 to 21 years) treated according to pediatric and adult protocols. Pediatric protocols contained a higher cumulative dose of anthracyclines (median, 305 mg/m^2^ vs. 202 mg/m^2^, *p* = 0.003). AYA treated with pediatric protocols presented higher 2-year incidence TRM in pediatric centers (23.5 ± 6% vs. 10.1 ± 3.2%, *p* = 0.046), and lower 5-year incidence of relapse or progression (33.3 ± 6.7% vs. 56.2 ± 5.3%, *p* = 0.002) [10]. In another study by Wennstrom et al. from the NOPHO group, no differences were found in the outcome of AML patients aged 10 to 30 years treated according to pediatric compared to adult protocol (OS, 60% vs. 65%) [11]. It appears that neither adult nor pediatric chemotherapy protocols are ideal for the needs of patients with AYA. More intensive pediatric regimens improve treatment outcomes by reducing the relapse rate, their use is associated with higher mortality due to treatment toxicity, as was also shown in the adolescent group in our study, although the differences between age groups were not statistically significant.

The presented study has some limitations. First, the small number of patients in the adolescent group (14% of all patients in the study) led to the analysis not always achieving statistical significance. Furthermore, when presenting the results of the treatment of adolescents, we showed only a narrow subgroup of all AYAs, which in total constitutes an extremely heterogeneous group of patients. It seems that only cooperation between pediatric and adult oncologists in clinical trials is a chance to learn the true picture of AYA and improve the treatment results of this group of patients.

## 5. Conclusions

To conclude, although we did not find age above 15 years as a poor survival indicator, age is still an important prognostic factor. Analysis of the treatment outcomes of patients with AML between strictly defined age intervals could give us more insight into the specific age at which the survival of young patients begins to deteriorate. Our results suggest that it is not adolescent age, although the treatment outcomes for this group of patients are still not satisfactory and treatment-related death rates are high. A personalized approach based on genetic profiling and adjusting the intensity of therapies to individual genetic risk, measurable residual disease assessment along with targeted therapies, and the introduction of new drugs could give the chance of further improvement in survival in adolescents.

## Figures and Tables

**Figure 1 cancers-16-01145-f001:**
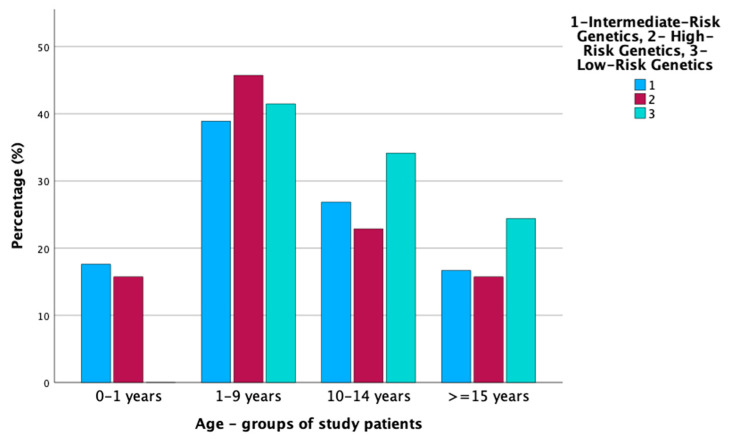
Genotype of leukemic cells according to the age of patients.

**Figure 2 cancers-16-01145-f002:**
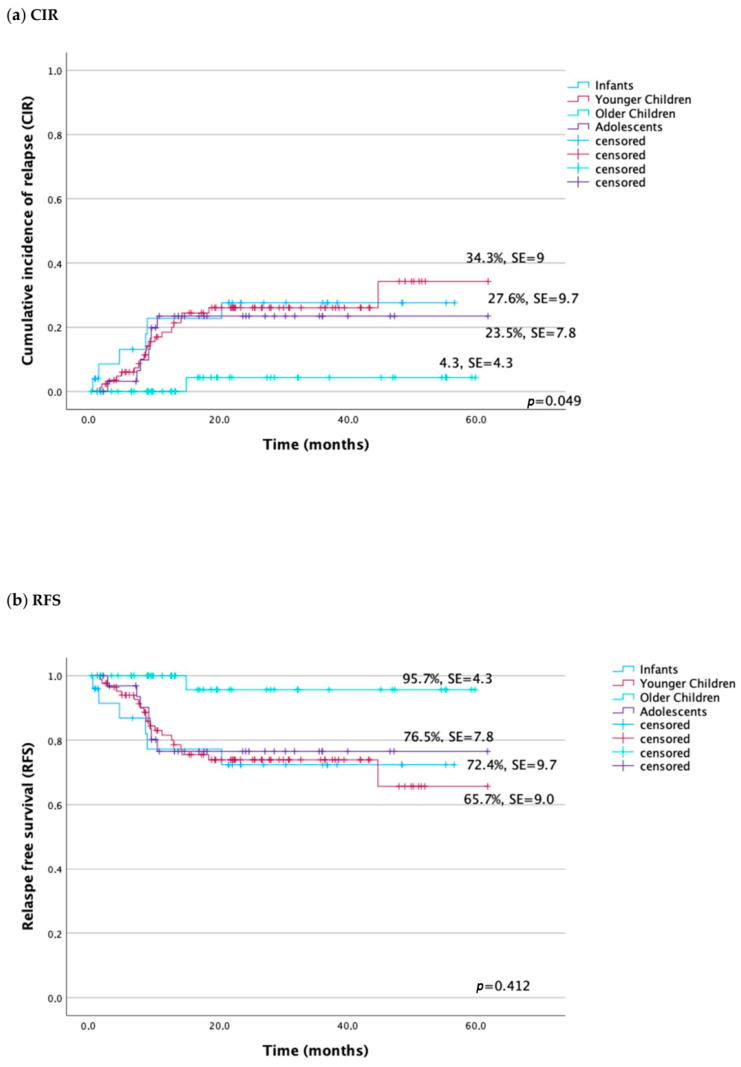
Probability of relapse in children and adolescents with AML according to age groups: (**a**) cumulative incidence of relapse (CIR), (**b**) relapse-free survival (RFS).

**Figure 3 cancers-16-01145-f003:**
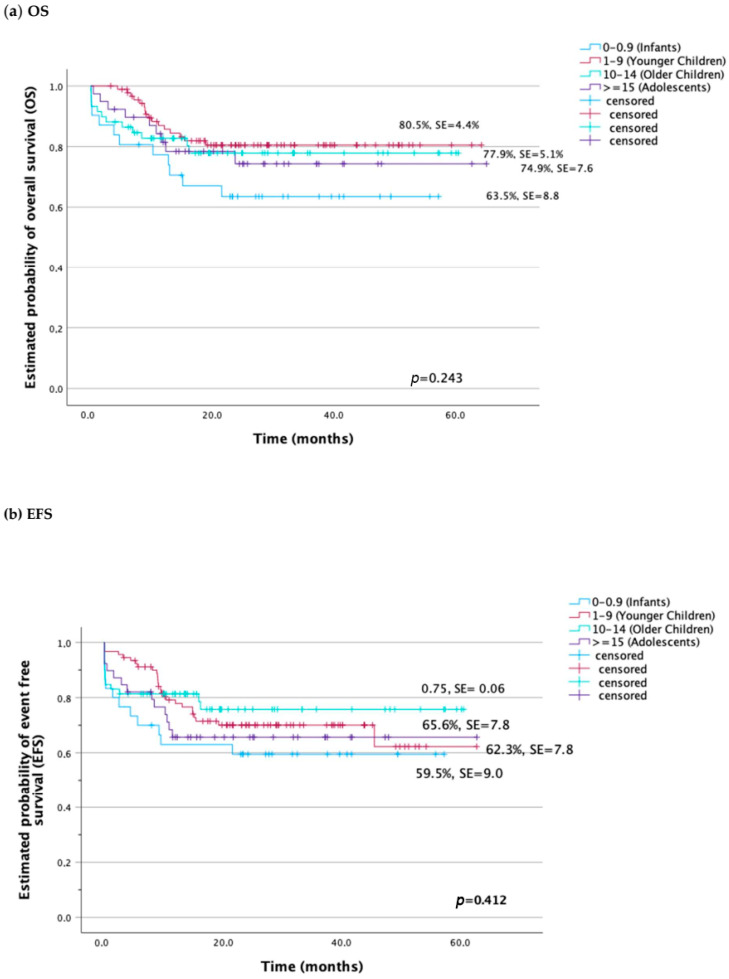
Probability of survival in children and adolescents with AML according to age groups: (**a**) overall survival (OS), (**b**) event-free survival (EFS).

**Table 1 cancers-16-01145-t001:** Comparison of patient characteristics among the age groups.

Feature	0–0.9 Years n = 31	1–9.9 Years n = 91	10–14.9 Years n = 59	Above 15 Years n = 39	*p*
Median age (range)	0.48 (0.09–0.92)	3.88 (1.0–9.5)	12.9 (9.5–14.9)	16.4 (15.1–17.8)	
**Sex**					NS
Female	11 (35.5%)	50 (55%)	32 (54%)	25 (64%)	
Male	20 (64.5%)	41 (45%)	27 (46%)	14 (36%)	
**Treatment protocol**					NS
AML-BFM 2012 Registry	23 (74%)	53 (58%)	32 (54%)	23 (59%)	
AML-BFM 2019	8 (26%)	38 (42%)	27 (46%)	16 (41%)	
**FAB classification**					<0.001
M0	1 (3.2%)	5 (5.5%)	1 (1.7%)	1 (2.6%)	
M1	1 (3.2%)	11 (12.1%)	8 (13.6%)	3 (7.7%)	
M2	0	24 (26.4%)	23 (39%)	12 (30.8%)	
M4	3 (9.7%)	5 (5.5%)	9 (15.3%)	12 (30.8%)	
M5	16 (51.6%)	29 (31.9%)	18 (30.5%)	11 (28.2%)	
M6	1 (3.2%)	0	0	0	
M7	6 (19.4%)	16 (17.6%)	0	0	
SG	3 (9.7%)	1 (1.1%)	0	0	
**Risk group**					0.012
SRG	0	21 (23%)	17 (29%)	15 (39%)	
IRG	15 (48%)	35 (38.5%)	18 (30.5%)	11 (28%)	
HRG	16 (52%)	35 (38.5%)	24 (40.5%)	13 (33%)	
**WBC at diagnosis (/μL)**					NS
<10,000	10 (32%)	27 (30%)	16 (27%)	9 (23%)	
10,000–50,000	9 (29%)	34 (37%)	21 (36%)	10 (26%)	
50,000–100,000	3 (10%)	10 (11%)	4 (7%)	6 (15%)	
>100,000	7 (22.5%)	13 (14%)	16 (27%)	10 (26%)	
ND	2 (6.5%)	7 (8%)	2 (3%)	4 (10%)	
**Genetic abnormalities**					
**Low-risk genetics**	0	19 (21%)	12 (20%)	15 (38%)	0.010
inv(16)(p13;q22)/CBFB::MYH11	0	3 (3.5%)	5 (9%)	10 (26%)	
t(8;21)(q22;q22)/RUNX1::RUNX1T1	0	16 (18%)	7 (13%)	5 (13%)	
**High-risk genetics**	13 (42%)	31 (34%)	14 (26%)	12 (31%)	0.005
FLT3-ITD and WT1	0	2 (2.5%)	4 (7%)	3 (7.5%)	
Monosomy-7	1 (3.2%)	0	2 (4%)	1 (2.5%)	
(9;22)(q24;q11.2)/BCR::ABL1	1 (3.2%)	1 (1%)	0	0	
(6;11)(q27;q23)/KMT2A::AF6	0	1 (1%)	0	2 (5%)	
(10;11)(p12;q23)/KMT2A::MLLT10	7 (22.5%)	8 (9%)	3 (5%)	3 (7.5%)	
(6;9)(p23;q34)/DEK::NUP214	0	1 (1%)	3 (5%)	0	
Complex karyotype	4 (13%)	18 (20%)	2 (4%)	3 (7.5%)	
**Intermediate-risk genetics**	18 (58%)	41 (45%)	29 (54%)	12 (31%)	NS
Normal karyotype	6 (19%)	16 (18%)	13 (24%)	7 (18%)	
Other	12 (39%)	25 (28%)	16 (29%)	5 (13%)	
ND of genetic analysis	0	1	4	0	
**HSCT in I CR**					
Yes	10 (32%)	31 (34%)	16 (27%)	11 (28%)	NS

SRG—standard-risk group, IRG—intermediate-risk group, HRG—high-risk group, WBC—white blood cells, HSCT = hematopoietic stem cells transplantation, CR—complete remission, ND—no data, NS—nonsignificant.

**Table 2 cancers-16-01145-t002:** Treatment outcomes among the age group.

Feature	0–0.9 Years n = 31	1–9.9 Years n = 91	10–14.9 Years n = 59	Above 15 Years n = 39	*p*
Probability of 5-year OS ± SD	63.5 ± 8.8	80.5 ± 4.4	77.9 ± 5.1	74.3 ± 7.6	NS
Probability of 5-year EFS ± SD	59.5 ± 9.0	62.3 ± 8.6	75.7 ± 6.1	65.6 ± 7.8	NS
Probability of 5-year RFS ± SD	72.4 ± 9.7	65.7 ± 9.0	95.7 ± 4.3	76.5 ± 7.8	0.049
Probability of 5-year CIR ± SD	27.6 ± 9.7	34.3 ± 9.0	4.3 ± 4.3	23.5 ± 7.8	0.049
CR (%)	27 (87)	87 (95)	53 (89.8)	35 (89.7)	0.015
NR (%)	0	4 (4.4)	1 (1.6)	3 (7.6)	
Relapse (%)	6 (19.4)	19 (20.9)	1 (1.6)	7 (17.9)	
Deaths (in total) (%)	11 (35.5)	16 (17.6)	12 (20.3)	9 (23.1)	NS
Early deaths (%)	4 (12.9)	0	5 (8.4)	1 (2.6)	0.039
<15 days	3 (9.7)	0	3 (5.1)	1	
15–42 days	1 (3.2)	0	2 (3.4)	0	
TRM (%)	2 (6.5)	4 (4.4)	3 (5.1)	4 (10.3)	NS
Death in progression (%)	7 (22.5)	12 (13)	4 (6.7)	4 (10.3)	NS

**Table 3 cancers-16-01145-t003:** Risk factor analysis for OS (overall survival) and EFS (event-free survival).

Features		Univariate Analysis		Multivariate Analysis	
		HR (95% CI)	*p*	HR (95% CI)	*p*
OS					
Age	<15 years	1		1	
	>15 years	1 (0.5–2.2)	NS	1.5 (0.7–3.2)	NS
Risk group	SRG	1		1	
	IRG	3.7 (1.0–12.6)	0.038	3.8 (1.1–13.3)	0.03
	HRG	6.3 (1.9–21.0)	0.002	6.3 (1.6–23.7)	0.006
Genotype	Non-high risk	1		1	
	High risk	2.0 (1.1–3.6)	0.014	1.8 (1.0–3.3)	0.046
WBC at diagnosis (/μL)	<100,000	1		1	
	≥100,000	2.4 (1.3–4.6)	0.004	2.4 (1.3–4.6)	0.004
EFS					
Age	<15 years	1		1	
	>15 years	1.2 (0.6–2.2)	0.490	1.8 (0.9–3.4)	NS
Risk group	SRG	1		1	
	IRG	2.8 (1.1–7.0)	0.021	3.3 (1.3–8.2)	0.01
	HRG	4.0 (1.6–9.5)	0.002	5.6 (2–15.5)	<0.001
Genotype	Non-high risk	1		1	
	High risk	1.1 (0.8–2.4)	NS	1.3 (0.7–2.2)	NS
WBC at diagnosis (/μL)	<100,000	1		1	
	≥100,000	2.1 (1.2–3.6)	0.007	2 (1.2–3.5)	0.009

HR—hazard ratio, WBC—white blood cells, non-high-risk genotype—low or intermediate-risk genotype, SRG—standard-risk group, IRG—intermediate-risk group, HRG—high-risk group, NS—nonsignificant

## Data Availability

The data are not publicly available due to privacy and ethical restrictions.

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
