# Peer review of "Treatment Outcomes of Adolescents Compared to Younger Pediatric Patients with Acute Myeloid Leukemia: Do They Need a Special Approach?"

_cancers, 2024, doi:10.3390/cancers16061145_

Round 1
Reviewer 1 Report
Comments and Suggestions for Authors
The authors in this retrospectively study have highlighted the role of characteristics and treatment outcomes of adolescents with AML compared to children and infants. Although it is not a novel work but will add to the literature from their part of the world.
The authors need to work on the mentioned comments to make this manuscript comprehensive and publishable in the journal of Cancers.
1. Manuscript needs English and Scientific language check.
2. The graphs (Kaplan Meier curves) needs better presentation and resolution.
3. Authors need to reduce the similarity of the manuscript text( plagiarism) to text less 20%.
4. The better study should have been the prospective.
5. Why is high leukocyte count above 100 000/ul at diagnosis unfavorable and opposite to the major similar studies done?
Comments on the Quality of English Language
Manuscript needs English and Scientific language check.
Author Response
Thank you for reviewing the manuscript and providing very useful comments and suggestions.
Ad. 1. English and Scientific language of the manuscript was carefully checked and improved.
Ad. 2. We also corrected the KM curves as suggested.
Ad. 3. Unfortunately, we used some part of the text from our previous paper to describe some general definitions of treatment outcomes, primary endpoints, which probably dangerously increased the percentage of similarities in our manuscript with others. We changed it, of course. Sorry about that. We carefully reviewed the paper in this regard.
Ad. 4. We totally agree with the Reviewer that prospective studies have a better value and would be the better way to answer the question of whether the adolescent needs a different treatment approach. The results of retrospective studies are often the basis for opening new prospective studies. We hope it will be possible in the nearest future.
Ad. 5. Thank you for this remark. According to our statistical analyses the high leukemic count (above 100 000/ul) was unfavorable risk factor for our patients. It was shown in univariate and multivariate tests – data shown, and also in KM analyses (data not shown). Only hyperleukocytosis compared to other values of leucocytes values (<10 000 vs >= 10 000; <50 000 vs. >= 50 000/ul) significantly influences EFS, RFS, OS. In the Discussion Section, we compared our results to the studies with similar results in this respect (Ref. 17 and 22). We added into the Discussion section results from other studies (by Creutzig et al. - Ref. 18.) in which the WBC count had no prognostic value.
Lines 351-356: However, the results of the analysis of the pediatric trials AML-BFM 93/98 and the adult trials AMLCG92/99 and AMLSG HD93/98A showed that only cytogenetic abnormalities, blast count after induction (above 5%) and age groups (above 2 years and less than 21 years) remained of prognostic significance. Leucocyte count above 100 000/ml at diagnosis didn’t have prognostic value [18].
Reviewer 2 Report
Comments and Suggestions for Authors
Authors analysed data and outcome from patients aged from 0 to 18 years old.
Few Comments
1/ Patients were classified in three genetic groups (Low Risk, High risk or Intermediate). As mentionned in Table S1, risk group were defined by the AML-BFM 2012 and later by the AML-BFM 2019. In the text (line 140,...) and in the table 1, the definition of risk group seems to be quite different --> Low Risk = Only CBF Leukemia,... Is there any data about NPM1 mutations,etc,..
2/ Legend of Figure 1 is subject of controversial - The three groups are Low-Risk/Intermediate Risk and High Risk. The use of the words"genotype without known clinical impact" is quite controversial.
3/ In the discussion, line 357 : "second best result" --> clearly behind older children and nearly from others.
General comment:
Results are difficult to interprate to conclude about higher TRM in AYA (Results are not significant between groups). Genetic findings seem to provide sensitivity to chemotherapy, but it will be more confortable to give this conclusion with more classical genetic data.
Author Response
Thank you for reviewing the manuscript and providing very useful comments and suggestions.
Ad. 1. Thank you for this comment. We decided to use genetic groups to see if genetics is the possible factor responsible for differences in treatment outcomes between age groups in our study (as suggested by the results of other studies). Although we detected NPM1 in our patients (10 patients), and CEBPA (6 patients) – all had some abnormalities in karyotype, so we classified them into the intermediate-risk genetics group and, as a result, only CBF patients were classified as low-risk genetics patients.
Ad. 2. We changed the description and used the nomenclature of three genetic groups as used in the main text.
Ad. 3. We changed this sentence to: “Favorable genetics could have contributed to the low relapse rate in adolescents compared to other age groups (5 year CIR: 23.5%±7.8, p=0.049).” lines 356-357
Yes, indeed, the differences in TRM between age groups were not statistically significant, although in adolescents it reached 10.3% (compared to 6.5 % in infants vs 4.4% in younger children vs 5.1% in older children), so we concluded based on the trend and the highest proportion, however, with the caution that the results are not significant. We highlighted it in the Results and Discussion sections.
Lines 305-307: We found that adolescents compared to children aged 1 to 14 years had a similar EFS rate, with lower OS, which could be explained by a higher incidence of TMR (10.3%), although the latter didn’t reach statistical significance.
Lines 390-393: “More intensive pediatric regimens improve treatment outcomes by reducing the relapse rate, their use is associated with higher mortality due to treatment toxicity, as was also shown in the adolescent group in our study, although the differences between age groups were not statistically significant.”
As stated in the response to the first remark (Ad 1) we divided patients into genetic groups according to the results of the genetics they presented, as described in the Method section. Conclusions are made based on statistical analysis of these data.
Round 2
Reviewer 2 Report
Comments and Suggestions for Authors
Thanks for the changes. It improves the quality of the message.